# Isolation and Characterization of Strain *Exiguobacterium* sp. KRL4, a Producer of Bioactive Secondary Metabolites from a Tibetan Glacier

**DOI:** 10.3390/microorganisms9050890

**Published:** 2021-04-21

**Authors:** Pietro Tedesco, Fortunato Palma Esposito, Antonio Masino, Giovanni Andrea Vitale, Emiliana Tortorella, Annarita Poli, Barbara Nicolaus, Leonardo Joaquim van Zyl, Marla Trindade, Donatella de Pascale

**Affiliations:** 1Institute of Protein Biochemistry, National Research Council, Via Pietro Castellino, 80131 Naples, Italy; pietro.tedesco@szn.it (P.T.); fortunato.palmaesposito@szn.it (F.P.E.); antonio.masino@unina.it (A.M.); giovanniandreavitale@gmail.com (G.A.V.); emiliana.tortorella@gmail.com (E.T.); 2Stazione Zoologica Anton Dohrn, Villa Comunale, 80122 Naples, Italy; 3Department of Biology, University of Naples Federico II, Via Cinthia, 80126 Naples, Italy; 4Institute of Biomolecular Chemistry, National Research Council, Pozzuoli, 80078 Naples, Italy; apoli@icb.cnr.it (A.P.); bnicolaus@icb.cnr.it (B.N.); 5Institute for Microbial Biotechnology and Metagenomics (IMBM), University of the Western Cape, Bellville, 7535 Cape Town, South Africa; vanzyllj@gmail.com (L.J.v.Z.); ituffin@uwc.ac.za (M.T.)

**Keywords:** *Exiguobacterium*, cold-environment, biosynthetic gene cluster, genome mining, bioactivities, anthelmintic, carotenoid, terpene

## Abstract

Extremophilic microorganisms represent a unique source of novel natural products. Among them, cold adapted bacteria and particularly alpine microorganisms are still underexplored. Here, we describe the isolation and characterization of a novel Gram-positive, aerobic rod-shaped alpine bacterium (KRL4), isolated from sediments from the Karuola glacier in Tibet, China. Complete phenotypic analysis was performed revealing the great adaptability of the strain to a wide range of temperatures (5–40 °C), pHs (5.5–8.5), and salinities (0–15% *w*/*v* NaCl). Genome sequencing identified KRL4 as a member of the placeholder genus *Exiguobacterium*_A and annotation revealed that only half of the protein-encoding genes (1522 of 3079) could be assigned a putative function. An analysis of the secondary metabolite clusters revealed the presence of two uncharacterized phytoene synthase containing pathways and a novel siderophore pathway. Biological assays confirmed that the strain produces molecules with antioxidant and siderophore activities. Furthermore, intracellular extracts showed nematocidal activity towards *C. elegans*, suggesting that strain KRL4 is a source of anthelmintic compounds.

## 1. Introduction

Natural products (NPs) represent the richest source of novel molecular scaffolds and chemistry and therefore they remain the best source of drugs and drug leads, but they also have other applications [1,2,3]. Microbial natural products have several advantages favoring their consideration in drug discovery and development as they can be produced by large-scale fermentation and the producer microorganisms can be engineered to overproduce the desired natural products and could thereby contribute to solving the supply bottleneck. Furthermore, metabolic pathway engineering can easily produce natural product analogues [2,4]. The vast untapped ecological biodiversity of microbes holds great promise for the discovery of novel natural products, thereby improving the odds of finding novel drug leads. It is estimated that only 1% of the microbial community has been cultivated in laboratories, implying that the majority of microbial natural products remains hidden [5]. The exploration of these extreme environments have led to the discovery of bacterial communities that have evolved novel bioactive compounds through their physiological adaptations to environmental stressors. However, due to their location and characteristics, extremophilic organisms are difficult to access and study. Among extreme environments, cold environments (< 5 °C) are the most widespread on our planet, but so far, applied research on cold-adapted microorganisms has mostly been based on the isolation of cold-adapted enzymes [6,7]. However, recent investigations have demonstrated that psychrophilic bacteria may be a promising source of NPs with anti-infective properties against MDR pathogens. These studies have shown that bacteria from Antarctica belonging to the *Pseudoalteromonas* and *Pseudomonas* genera, produce a variety of bioactive compounds able to inhibit the growth of different pathogens [8,9,10] The deep sea represents another cold “niche” that is the subject of a growing interest in the last few years and has yielded a significant number of NPs with different bioactivities [11,12,13]. By contrast, alpine habitats are at the very beginning of exploration and bioprospecting for NPs producers [14,15], and a few recent studies have highlighted the high microbial diversity present in these extreme habitats, encouraging the isolation and exploitation of alpine bacteria for biotechnological purposes [16]. Several studies reported the bioremediation potential of alpine microorganisms, their antioxidant, photoprotective activities through the production of pigments [17,18] and the potential for NPs discovery [19,20]. The Qinghai-Tibet Plateau, often called the ‘world’s roof’ or ‘the third pole’ is located in the southwest of China and is the highest and largest region with permafrost in the world. These conditions make it a unique alpine ecosystem, sensitive to changes in climate and surface conditions [21,22]. In this paper, we characterized an alpine bacterium, strain KRL4, isolated from Karuola Glacier sediments, which displays a range of bioactivities which includes anthelminthic activity. Genome sequencing and mining for biosynthetic gene clusters did not yield obvious clues as to the nature of the anthelminthic activity, although several bioactivity assays confirm the strain’s potential for future drug discovery applications.

## 2. Materials and Methods

### 2.1. Isolation of the Strain and Characterization

KRL4 was isolated from a glacier sample (composed by a mixture of ice and soil) collected from Karuola glacier located on the Tibetan Plateau, 5200 m above sea level, China (geographical coordinates: 90°19.23′ E, 28°90.71′ N). Strain KRL4 was isolated at 4 °C on PYG agar plates containing peptone 5.0 g/L, yeast extract 4.0 g/L, glucose 1.0 g/L, CaCl_2_ 0.2 g/L, MgSO_4_ 7H_2_O 0.4 g/L, K_2_HPO_4_ 1.0 g/L, KH_2_PO_4_ 1.0 g/L, NaHCO_3_10.0 g/L NaCl 2.0 g/L, and 17 g/L agar. After isolation KRL4 was routinely cultured on LB medium (5 g/L yeast Extract, 10 g/L tryptone, 10 g/L NaCl) or M9 salts medium (M9 salts: Na_2_HPO_4_ × 12H_2_O 17.4 g/L, KH_2_PO_4_ 3.02 g/L, NaCl 0.51 g/L, NH_4_Cl 2.04 g/L. Trace metals salts: Na_2_EDTA × 2H2O 15 mg/L, ZnSO_4_ × 7H_2_O 4.5 mg/L, CoCl_2_ × 6H_2_O 0.3 mg/L, MnCl_2_ × 4H_2_O 1 mg/L, H_3_BO_3_ 1 mg/L, Na_2_MoO_4_ × 2H_2_O 0.4 mg/L, FeSO_4_ × 7H_2_O 3 mg/L, CuSO_4_ × 5H_2_O 0.3 mg/L. MgSO_4_ 0.5 g/L, CaCl_2_ 4.38 mg/L) supplemented with glucose as sole carbon source at 3 g/L or TYP (15 g/L tryptone, 15 g/L yeast extract, 10 g/L NaCl). The pH dependence of the strain was evaluated by growing the strain in liquid minimal medium M9 buffered at different pH (5.5–8.5) using glucose as sole carbon source. Temperature dependency was evaluated by growing the strain in liquid M9 medium supplemented with glucose at different temperature ranging from 5 to 40 °C. Carbon-source assimilation tests were carried out in M9 minimal medium supplemented with different carbon sources at the concentration of 3 g/L. For salt dependencies, the strain was cultured in M9 minimal medium with varying NaCl concentrations (0 to 20% *w*/*v*). All the cultures were performed in duplicate or triplicate in shake-flasks at 20 °C and 200 rpm, in a volume of 250 mL. Growth was monitored by optical density at 600 nm using a Cary100 spectrophotometer (Mettler Toledo, Columbus, OH, USA). Imaging of the bacterium was conducted by transmission electron microscopy (TEM). 2μL of bacterial culture were diluted in PBS containing 2% Glutaraldehyde (final 10 μL) to fix them. The sample was applied to formvar 100-mesh grids and incubated for 10 min. Grids were washed twice with filtered distilled water and stained using 1.5% UA in water for 10 min. Afterwards, they were washed with water to remove the excess staining solution and grids were air-dried. Images were acquired from grids using a FEI Tecnai 12 transmission electron microscope (FEI Company, Hillsboro, OR, USA) equipped with a Veleta CCD digital camera (Olympus Soft Imaging Solutions GmbH, Münster, Germany) and operating at 120 kV. Images were collected at magnifications of ×30000 and ×49000. The strain has been deposited within the “Centre de Ressources Biologiques de l’Institute Pasteur (CRBIP), under the accession number CIP 111565.

### 2.2. Bacterial Identification

A KRL4 colony cultured overnight at 20 °C on LB agar was suspended in 25 μL of sterile distilled water, heated to 95 °C for 10 min, and cooled on ice for 5 min. Then, 2 μL of each cell lysate was used for the amplification via PCR of 16S rRNA genes using universal primers 27F (5′-AGA GTT TGA TCM TGG CTC AG-3′) and P6 (5′-CGG TTA CCT TGT TAC GAC TT-3′) [23]. Direct sequencing was performed on both DNA strands using an ABI PRISM 310 Genetic Analyzer (Applied Biosystems, Forster City, CA, USA).

### 2.3. Genome Sequencing and Analysis.

Strain *Exiguobacterium* sp. KRL4 was cultured in LB broth at 20 °C for 48 h. High-quality genomic DNA was extracted and purified using a GenElute™ Bacterial Genomic DNA Kit (Sigma, St Louis, MO, USA). The genomic DNA was purified according to the manufacturer’s instruction for extraction of nucleic acid from Gram-positive bacteria. The DNA was further purified using the Qiagen gel extraction kit (Qiaex II; catalog no. 20021; Qiagen, Hilden, Germany) for preparation of sequencing libraries. The size, purity, and DNA concentration of genomic DNA was determined by agarose gel electrophoresis, ratio of absorbance values at 260 nm and 280 by nanodrop (Thermo Scientific, Waltham, MA, USA). Sequence libraries were prepared with the Illumina Nextera XT library prep kit and sequenced with using a MiSeq reagent kit V3 (Illumina, San Diego, CA, USA). A 10% phiX V3 spike was included during sequencing as per the manufacturer’s instructions (Illumina Nextera XT guide). Sequencing resulted in 2x (forward and reverse sequencing) 300-bp reads, performed on the Illumina MiSeqGenome assembly was performed using CLC Genomics Workbench version 7.5.1 employing a length fraction of 0.7 and similarity fraction of 0.9. Contigs smaller than 500 bp were removed from the final assembly. The final number of contigs was 48. The draft genome sequence was deposited in the GenBank database under the accession number MOLV00000000.1. The project information and its association with MIGS version 2.0 compliance are summarized in Appendix A. A basic annotation was performed using the RAST server [24]. The coding sequences were translated and used to interrogate NCBI non-redundant database, such as Kyoto Encyclopedia of Genes and Genomes (KEGG), and Clusters of Orthologous Groups of proteins (COGs) [25] databases. These data sources were combined to assert a product description for each predicted protein. Non-coding genes, and miscellaneous features were predicted with Bacterial Annotation System (BASys) [26], Protein Family database (PFAM) 3.0 [27], and Signal 4.1 [28]. The genome taxonomy database toolkit (GTDB-Tk v1.1.0) was used to establish a whole genome phylogeny for KRL4 [29]. The software antiSMASH 5.1.1 was used to identify potential secondary metabolites clusters [30] and Biosynthetic Gene Similarity Clustering and Prospecting Engine (BiG-SCAPE) was used to establish similarity among pathways [31]. For pathway comparisons among the Exiguobacteria, the genbank annotations were retrieved for 78 RefSeq genomes as of 27-03-2021. This dataset contained 43 *Exiguobacterium* and 35 *Exiguobacterium_A* genomes including the type species of each species in the GTDB phylogeny. Average nucleotide and amino acid identities were calculated using the Kostas lab tools [32]. Prophage genomes were identified using the PHAST server [33]. BAGEL and antiSMASH were used to identify possible bacteriocins [34].

### 2.4. Membrane Composition Analysis

KRL4 lipid extracts were obtained from freeze-dried cells harvested at the stationary growth phase after culturing at a temperature of 20 °C for 5 days in TYP medium. Quinones were extracted from freeze-dried cells with *n*-hexane and were purified by thin layer chromatography (TLC) on silica gel (0.25 mm, F254, Merck Darmstadt, Germany) eluted with *n*-hexane/ethyl acetate (96:4, by vol.). The purified UV-bands from TLC were then analyzed by LC/MS on a reverse-phase RP-18 Lichrospher column eluted with *n*-hexane/ethylacetate (99:1, by vol.) with a flow rate of 1.0 mL min^−1^ and identified by Electrospray Ionization Mass Spectrometry (ESI/MS) and protonic Nuclear Magnetic Resonance (^1^H-NMR) spectra. NMR spectra, recorded at the NMR Service of Institute of Biomolecular Chemistry of CNR (Pozzuoli, Italy), were acquired on a Bruker DPX-400 operating at 400 MHz, using a dual probe. For complex lipid extraction, the residual cellular pellet, after *n*-hexane extraction, was subjected to a following extraction with CHCl_3_/MeOH/H_2_O (65:25:4, by vol.) for polar lipids recovery. The polar lipid extracts were analyzed by TLC on silica gel (0.25 mm, F254, Merck) eluted in the first dimension with CHCl_3_/MeOH/H_2_O (65:25:4, by vol.) and in the second dimension with CHCl_3_/MeOH/Acetic acid/H_2_O (80:12:15:4, by vol.). All polar lipids were detected by spraying the plates with 0.1 % (*w*/*v*) Ce(SO_4_)_2_ in 1M H_2_SO_4_ or with 3% (*w*/*v*) methanolic solution of molybdophosphoric acid followed by heating at 100 °C for 5 min. Phospholipids and aminolipids were detected by spraying the TLC plate with the Dittmer–Lester and the ninhydrin reagents, respectively, and glycolipids were visualized with α-naphthol. Fatty acid methyl esters (FAMEs) were obtained from complex lipids by acid methanolysis and analyzed using a Hewlett Packard 5890A gas chromatographer (Hewlett Packar, Palo Alto, CA, USA) fitted with an FID detector and equipped with an HP-V column with a flow rate of 45 mL/min using the temperature program of 120 °C (1 min), from 120 to 250 °C at 2 °C/min. The identification of the compounds was made with standards and by interpretation of mass spectra.

### 2.5. Bioactivity Assays

#### 2.5.1. Preparation of Crude Extracts

A single colony of strain KRL4 was used to inoculate 3 mL of liquid TYP or PYG media in sterile bacteriological tubes. After 48 h of incubation at 20 °C at 200 rpm this pre-inoculum was used to inoculate 100 mL of TYP medium or PYG in a 500 mL flask, at an initial cell concentration of 0.01-OD600/mL. The flasks were incubated up to 5 days at 20 °C at 220 rpm. The cultures were then centrifuged at 6800 *g* at 4 °C for 30 min, and the exhausted culture broths were collected and stored at −20 °C. The exhausted culture broths were subjected to organic extraction. Extractions were performed using 3 volume of ethyl acetate in a 500 mL separator funnel. Extractions with the resin HP20 were also performed. For the resin extractions, the samples were incubated with the resin (5 g resin/ 100 mL broth), which was then collected after 4 h of incubation at room temperature, washed with distilled water and eluted with methanol. To obtain intracellular extract, the collected pellet was subjected to sonication and then centrifugation at 10000 *g* at 4 °C for 30 min to remove cells debris. The resulting supernatant was processed similarly as the exhausted culture broth. The organic phase collected was evaporated using a Laborota 4000 rotary evaporator (Heidolph, Schwabach, Germany), and the extracts were weighed, dissolved in 100% DMSO at 50 or 10 mg/mL and stored at −20 °C.

#### 2.5.2. Minimal Inhibitory Concentration Assay (MIC)

To evaluate the antimicrobial potential of KRL4, extracts were placed into each well of a 96-well microtiter plate at an initial concentration of 2% (*v*/*v*) and serially diluted using LB medium. Wells containing no compound represented the negative control. DMSO was used as control to determine the effect of solvent on cell growth. A single colony of a pathogen was used to inoculate 3 mL of liquid LB media in a sterile bacteriological tube. After 6–8 h of incubation, growth was measured by monitoring the absorbance at 600 nm and about 4 × 10^5^ CFU were dispensed in each well of the prepared plate. Plates were incubated at 37 °C for 24h and growth was measured with a Cytation3 Plate Reader (Biotek, Winoosky, VT, USA) by monitoring the absorbance at 600 nm. To evaluate the antimicrobial capacity of the extracts, a panel of model MDR pathogens were used: *Burkholderia metallica* LMG 24068 [35], *K. pneumonie* DF12SA [36], *E. coli* ATCC 10536 [37], *Listeria monocytogenes* MB 677 [38], *Pseudomonas aeruginosa* PA01 [39], *Staphylococcus aureus* 6538P [40], *Staphylococcus epidermidis* ATCC 35984 [41]. All pathogens were cultured overnight in LB at 37 °C with aeration at 210 rpm.

#### 2.5.3. Nematode Liquid Toxicity Assay

The *C. elegans* strain N2 Bristol (wild type) was purchased from the *Caenorhabditis* Genetic Center (CGC), University of Michigan, USA. The nematodes were propagated on Nematode Growth Medium (NGM, 2.5 g/L peptone, 2.9 g/L NaCl, 17 g/L bacto-Agar, 1 mM CaCl_2_, 5 mg/L cholesterol, 3.40 g/L KH_2_PO_4_ and 0.12 g/L MgSO_4_), supplemented with *E. coli* OP50 as carbon source, and incubated at 20 °C [42]. To test the effect of crude extracts or compounds on *C. elegans* viability a liquid toxicity assay was performed in 24-well plates. Each well contained a 400 μL solution of M9 buffer, 5 μg/mL cholesterol, and *E. coli* OP50 at the concentration of 0.5 OD/mL as carbon source. Extracts or compounds at different concentration were then added to each well and 1% DMSO was also added as control to evaluate solvent effects on nematodes. *C. elegans* was synchronized by the bleaching treatment [42], and 30–40 L4 worms were transferred to each well and incubated at 20 °C up to seven days. The wells were scored for living worms every 24 h. A worm was considered dead when it no longer responded to touch. For statistical purposes, 3 replicates per trial were carried out with a unique egg preparation. Anthelmintic activity was reported as a function of the percentage of the number of alive worms in a well after the period of incubation with the extract, divided by the number of initial worms in that well.

#### 2.5.4. DPPH Radical Scavenging Assay

The capacity to scavenge the stable radical DPPH (Diphenyl-1-picrydrazyl) of all the extracts produced was assessed [43]. The extracts were dissolved in MeOH at an initial concentration of 50 mg/mL, and 4 µL of each extract were put in the first well, while the other wells were filled with 100 µL of the same solvent. Then, a 2-fold serial dilution starting from the first well was performed for all the extracts. Finally, the final volume of 200 µL was reached by adding 100 µL of a 0.2 mM methanolic solution of DPPH to all the wells. Ascorbic acid (AA) and MeOH were used respectively as positive and negative controls.

#### 2.5.5. Blue Agar CAS Assay for Siderophore Activity

To evaluate the ability of strain KRL4 to produce an active siderophore, a modified version of the blue agar CAS assay was performed. Briefly, blue dye solution prepared according to Louden et al. [44] and mixed in 1:10 ratio with LB agar solution and poured in sterile petri dish (9.6 cm diameter). Small holes with a diameter of 1 cm were cut in these dishes and 10 uL of KRL4 extracts were placed inside the holes. 10 µL of 10 mM EDTA was used as positive control, while 10 µL of DMSO was used as a negative control. The plates were then incubated at 4 °C in the dark for one night and then manually inspected. If an iron chelator such as a siderophore removes iron from the dye complex, the color of the agar changes from blue to orange.

## 3. Results and Discussion

### 3.1. Isolation and Characterization of Strain KRL4

Strain KRL4 was isolated from Karuola glacier sediments collected at 5200 m of altitude, after incubation on PYG medium at 4 °C for 15 days. After the first isolation, the bacterium grew on solid LB medium at 20 °C forming colonies after 2–3 days. Such colonies are 1–3 mm diameter, small, white, with a smooth border, and turn orange in 2 days. Cells have a rod shape, are non-motile and are 2.0–2.5 by 0.5 to 0.8 µm as shown in Figure 1. KRL4 was also able to grow in M9 + glucose liquid medium in a temperature range of 5 to 40 °C and pH of 5.5 to 8.5, with a temperature optimum of 30 °C and a broad pH optimum between 6.5 and 7.5. Growth was observed in concentrations ranging between 0 and 15% NaCl. The carbon source utilization was assessed by supplementation with sugars (D-glucose, xylose, D-maltose, fructose, lactose, D-sorbitol, sucrose, cellobiose). and structural amino acids (glycine, L-histidine, L-cysteine) (Table 1). These results are in agreement with what was reported for other strains of the genus, which are characterized by high versatility of growth conditions that allow them to colonize and thrive in very different ecological niches [45]. From 16S rRNA sequence analysis (see more in detail below) strain KRL4 could be assigned to genus *Exiguobacterium*. Members of this genus have shown a wide tolerance for pH (4–11) and temperatures (4–50 °C), even strains isolated from cold environments [46]. Interestingly, strain KRL4 showed higher than expected NaCl tolerance (up to 15%) for a non-marine strain. However, two different isolates from a Chilean altiplano and a Canadian freshwater habitat were reported to confer slightly lower, yet similar NaCl tolerance values [46,47]. This once again underlines the level of plasticity and adaptative capacity of strains belonging to this genus.

Chromatographic analysis of quinones revealed the presence of two abundant UV-absorbing bands (Appendix A). ^1^H-NMR spectrum showed that the quinones present are menaquinone (MK) types (Appendix A). ESI/MS analyses of the quinone content of KRL4 strain gave molecular peaks corresponding to MK6 (6 isoprenoid residues in the side chain) and MK7(7 isoprenoid residues in the side chain), 85% and 15%, respectively (Appendix A). 

This is contrary to most species in the genus *Exiguobacterium* where MK7 dominates [18]. The isolate KRL4 possessed complex lipids based on fatty acids. The total lipid content was about 10% of the total dry weight of cells cultured at 20 °C in LB medium. Under these conditions, four major phospholipids, 1,2 diacylglycero-3-phosphorylethanolamine (PEA), 1,2 dipalmitoyl-3-glycero-phosphatidic acid (AP), 1,3-di(3-sn-phosphatidyl)-sn-glycerol disodium salt (cardiolipin, CL), and 1-(3-sn-phosphatidyl)-sn-glycerol (PG) were identified (Appendix A). These are the major known polar lipids produced by *Exiguobacterium* species [48].

Four putative cardiolipin synthases were identified in the KRL4 genome. Although it was shown for *E. coli* that inactivation of a cardiolipin synthase improved freeze-thaw survival rates, increased cardiolipin synthase expression was one of the mechanisms by which *Planococcus halocryophilus* Or1 adapted to growth at low temperatures [49,50]. FAMEs’ composition, determined on cells cultured under standard conditions, was characterized by the abundance of branched acyl chains. The most abundant was C18:1 (40%), other components were iC15:0 (3.4%), aiC15:0 (0.6%), iC16:0 (2.0%), C16:1 (8.0%), C16:0 (5.9%), C17:1 (5.3%), iC17:0 (19%), aiC17:0 (2.3%), C17:0 (2.0%), C18:0 (6.3%), iC19:0 (0.5%) and aiC19:0 (0.5%) (Appendix A
Appendix A).

**Table 1 microorganisms-09-00890-t001:** Classification and general features of strain KRL4.

Property	Term	Evidence Code ^a^
Classification	Domain Bacteria	TAS [51]
	Phylum Firmicutes	TAS [52]
	Class Bacilli	TAS [48]
	Order Bacillales	TAS [53]
	Family Bacillales_Incertae Sedis XII	TAS [53]
	Genus Exiguobacterium_A	TAS [54]
	Positive	IDA
Gram stain	Rod	IDA
Cell shape	non-motile	IDA
Motility	Not reported	NAS
Sporulation	4–40 °C	IDA
Temperature range	30 °C	IDA
Optimum temperature	5.5–8.5; 6	IDA
pH range; Optimum	D-mannose, Dextrin, D-fructose, sucrose D-maltose, α-D-glucose, D-lactose, D-sorbitol, D-galattosio, Lactulose, Dextrine, glycerol L-hystidine, L-cysteine, Xylan	IDA
Carbon source	Soil, sediment	IDA
Habitat	0–15 % NaCl (*w*/*v*)	IDA
Salinity	Aerobic	IDA
Oxygen requirement	free-living	IDA
Biotic relationship	non-pathogen	NAS
Pathogenicity	China/Tibet	
Geographic location	August 2011	
Sample collection	90°19.23′ E	
Latitude	28°90.71′ N	
Longitude	5200 m	
Altitude		

^a^Evidence codes -IDA: Inferred from Direct Assay; TAS: Traceable Author Statement (i.e., a direct report exists in the literature); NAS: Non-traceable Author Statement (i.e., not directly observed for the living, isolated sample, but based on a generally accepted property for the species, or anecdotal evidence). These evidence codes are from the Gene Ontology project [55].

### 3.2. Genome Properties and Identification of Isolate KRL4

Genome sequencing and assembly resulted in a draft genome of 3,118,075 bp in size with a G+C content of 46.9% (Table 2), similar to the other genomes sequenced of *Exiguobacterium* members. A total of 3201 genes were annotated, 3079 are protein-coding genes and 79 are RNA genes and 43 pseudogenes were identified. It is worth noting that a high portion of genes (1557) were annotated as hypothetical. The classification of genes into COGs functional categories is shown in Appendix A.

Using the RDP classifier and Silva database search and classify option, the 16S rRNA could only be assigned to the genus *Exiguobacterium*. The partial (1412 bp) 16S rRNA sequence shared 99% sequence identity with *Exiguobacterium* species such as: *E. antarcticum* strain DSM 14480 (NR_043476.1), *Exiguobacterium soli* strain DVS 3y (NR_043241.1), *Exiguobacterium undae* strain DSM 14481 (NR_043477.1) and strain L2 (NR_114811.1). The genus *Exiguobacterium* consists of strains isolated from vastly different environments, including hot springs and Siberian permafrost [56]. This peculiarity is well reflected in the phylogenetic tree (Figure 2) which is principally divided into two clades, the upper one including mostly strains isolated from extreme cold environment, while the lower one encompassing mesophilic and thermophilic bacteria [56]. This division is also supported by the genome taxonomy database phylogeny of these bacteria (see below). Given its isolation source and growth characteristics, it is not surprising that strain KRL4 groups with other psychrophilic isolates. Interestingly strain KRL4 grouped together with *E. antarcticum* DSM 14480 and *E. soli* strain DVS 3y, two strains isolated from Antarctica, rather than *E. sibiricum* and *E. undae* from permafrost in Northeast Russia, which are geographically closer. However, the two Antarctic strains were isolated from glacier-like locations (frozen lake and moraine) and the similarity of the isolation micro-environment between the Antarctic strains and strain KRL4 could explain their genetic closeness.

Whole genome-based classification of KRL4 indicated that this isolate is a new species in the placeholder genus *Exiguobacterium*_A according the GTDB bacterial phylogeny (Appendix A). In this phylogeny, it is most closely related to *Exiguobacterium* _A *antarcticum*.

### 3.3. Genomic Insights of Strain KRL4

From genome analysis, three secondary metabolite clusters could be identified: two of them coding for terpenes, and one for a siderophore (Figure 3 and Appendix A). None of these clusters have been characterized previously, underlining the novelty of this strain and *Exiguobacteria* in general. One of the terpene clusters is clearly responsible for synthesis of a carotenoid giving these bacteria their characteristic color and occurs in representatives from both *Exiguobacterium* and *Exiguobacterium*_A genera. These molecules are 40 carbon intermediates of the carotenoid synthesis pathway. The characteristic enzyme of the pathway is phytoene synthase, which catalyzes the condensation of two molecules of geranylgeranyl diphosphate (GGPP) to give prephytoene diphosphate (PPPP) and the subsequent rearrangement of the cyclopropylcarbinyl intermediate to phytoene [61]. The pathway appears to be highly conserved among the Exiguobacteria, with the pathways from *Exiguobacterium* species being strongly associated with an additional α-amylase and transporter compared with those from *Exiguobacterium*_A (Figure 3A). Furthermore, the pathway from *E*_A *antarcticum* B7 appears to encode a unique phytoene desaturase. The second terpene related “pathway” only occurs in *Exiguobacterium*_A species and contains a phytoene synthase seemingly co-transcribed with flagella component genes which could be part of a phototactic response under control of a two-component regulatory system [62,63], possibly lacking in *Exiguobacterium* species.

No siderophore pathways could be identified in *Exiguobacterium* genomes, whereas 26 of the 35 *Exiguobacterium*_A genomes contained a siderophore pathway. The siderophore core cluster is composed of two *iucA/iucC* siderophore biosynthesis enzyme encoding geens, one acyl-CoA synthase and a putative acyl carrier protein. The compounds produced by these pathways are currently uncharacterized. The siderophore pathways from *Exiguobacterium*_A species split into two gene cluster families one of which occurs predominantly in *E*_A *acetylicum* and *E*_A *indicum* species and the other in *E*_A *undae*, *E*_A *antarcticum* and *E*_A *sibiricum* species (Figure 3B). The main difference is the association with FecCD family transporters which are likely responsible for ferric iron uptake [64] in these latter species and are unique to these pathways and are absent in other *Exiguobacterium*_A genomes altogether. KRL4 also possesses a suit of four petrobactin uptake related ABC transporters located distant from the siderophore pathway genes (BLD48_RS01685 to BLD48_RS01700).

A comparison of these KRL4 pathways with those identified in other *Exiguobacterium*_A species shows a high level of synteny and amino acid identity between the terpene and siderophore pathways (Figure 3B) suggesting strong selective pressure on these pathways, even though the strains originate from geographically distant environments.

The genome was further mined for putative antimicrobial peptide-encoding pathways. A total of ten RiPPs were identified indicating that these are rare among the *Exiguobacteraceae*
**(Figure 3**C). Of these, one pathway occurs in four *Exiguobacterium*_A genomes while the others all appear unique to the genomes they are found in, suggesting that individual *Exiguobacterium* isolates could yield highly novel peptide-based compounds albeit at low frequency. In KRL4 the C-terminal regions of two open reading frames on contig 19 showed low similarity (57% over 114 aa and 57% over 145 aa) to the Peptidase_M23 domain of Class III bacteriocin Zoocin A produced by *Streptococcus equi* [65], however, no immunity protein could be identified. Whether the products of these ORFs have antimicrobial activity is yet to be determined.

Another interesting feature many *Exiguobacteria* have is the presence of a proteorhodopsin protein which allows light-driven proton translocation to the extracellular milieu for energy generation through F_0_F_1_ ATPase [66]. The structure of several proteorhodopsins has been determined including that of *E. sibericum* 255-15 [67] and the protein from KRL4 shares 97% amino acid identity with it, and likely fulfills the same role in KRL4 harvesting extra energy by means of phototrophy [66]. Located directly upstream from the proteorhodopsin is an arsenic resistance operon. *Exiguobacterium* isolates SH31 and S17 were shown to be highly arsenic resistant [68]. Analysis of heavy metals in soils around the Tibetan Plateau showed high arsenic, chromium, nickel, zinc, and copper concentrations at the foot of the Karuola glacier [69]. This likely indicates why KRL4 has retained (or gained) the full arsenic resistance operon compared with *E. antarcticum* B7 and *E. sibericum* 255-15 which lack the *arsC*, *arsD* and *arsA* genes. The absence of these genes appears to be common in the case of most class I and III *Exiguobacterium* species and is likely indicative of the environment they find themselves in [68].

Moreover, three bacteriophage-related regions were identified on the KRL4 genome. Two of these together appeared to be an intact myovirus genome (±66kb), the proteins of which appear to be most closely related to *Paenibacillus* phage Tripp with the terminase large subunit showing 68% similarity. When using the prophage genome as search query and performing a BLASTn search against all *Exiguobacterium* genomes available on the NCBI database, it was only found on KRL4. When searching against the NCBInr database, only small fragments of this genome found matches in *Exiguobacterium* genomes, again indicating that at a nucleotide level, no closely related phages are currently described in the NCBI database. A nucleotide match could also not be found against the “earths” metavirome contig set [70]. This demonstrates that KRL4 harbors a highly unique bacteriophage. tBLASTx analysis did identify three contigs encoding proteins with amino acid similarity to the prophage on KRL4 from the “earths” virome. Two of these contigs are from a metagenome of ionic liquid and high solid enriched fermentations while the third is from *Arabidopsis* rhizosphere microbial communities [71].

### 3.4. Bioactivity Properties

To evaluate and confirm the biosynthetic potential of strain *Exiguobacterium_*A KRL4 from genome mining, several bioassays were performed.

Blue agar CAS assays were performed to evaluate the production of siderophores. The test was performed using the extract and not through culturing directly on the plate due to the toxicity of CAS for some Gram-positive bacteria [43]. After overnight incubation of the extracts in the agar wells, an orange halo was observed for extract generated from bacteria cultured in LB broth, but not TYP medium (Figure 4A). This confirms that KRL4 can produce a siderophore as suggested by genome analysis. Siderophores have received much attention in recent years because of their potential roles and applications in various areas of environmental research [72]. Thus far, different reports connected strains belonging to genus *Exiguobacterium* with the production of siderophore able to promote plant growth [73,74]. The radical scavenging activity of the carotenoid produced by the strain was evaluated through DPPH assays. Extracts exhibited radical scavenging at the highest concentration (1 mg/mL) which was independent of the culturing media used (Figure 4B). However, it is noticeable that the extract obtained in TYP-glucose media displayed the best scavenging activity, which was comparable with the positive control (ascorbic acid). It retained over 70% of activity at 0.125 mg/mL and 50% at 0.0625 mg/mL. *Exiguobacterium acetylicum* S01 has been reported to produce 6 different carotenoids (two of which were novel) and is considered endowed with promising antioxidant and anti-inflammatory activities [75].

Antimicrobial compounds produced by *Exiguobacterium* species have been described previously [76,77]. *Exiguobacterium mexicanum* produces two antimicrobial metabolites that show broad spectrum activity [77]. Neither of these are terpenoid or siderophore related compounds and the sequenced *E. mexicanum* genomes available do not give clues as to the origin of these compounds. KRL4 may harbor as of yet undiscovered secondary metabolite pathways especially as a large portion of the proteins are annotated as hypothetical proteins. Nevertheless, antimicrobial assays on our selected panel of pathogens did not show any significant activities from extracts generated. By contrast, the intracellular HP20 extract showed anthelmintic activity towards the nematode *C. elegans* (Figure 4C). Specifically, at 5 mg/mL concentration only 20% of the worm population was still alive after 3 days, while 37% of nematodes survived at 2.5 mg/mL suggesting a dose-response effect. Surprisingly, active extract was obtained only with HP20 resin and not with ethyl acetate extraction, denoting a polar nature of the anthelmintic compound. It is not clear which BGCs could be responsible for this activity. A possibility could be represented by the terpene clusters, as it is long known that a few terpenes isolated from plants have shown anthelmintic activities [78]. Further studies are required to identify the responsible molecules. To the best of our knowledge no anthelmintic activity has ever been reported from the *Exiguobacteraceae* and in general there are very few reports for bacterial-derived anthelmintic drugs. The principal problem is that anthelmintic drug discovery is not a priority for the pharmaceutical industry and there is a stronger focus on plant extracts rather than bacteria [79,80], despite the fact that the last anthelmintic drug (nemadectin) discovered was of a bacterial source [81]. Our findings underline the great potential of microorganisms isolated from extreme environments as sources of NPs.

## 4. Conclusions

Bioprospecting from extreme environments is considered one of the best strategies to identify new drugs. In this work, we isolated a new strain from a Tibetan glacier and determined the complete genome sequence of *Exiguobacterium sp.* KRL4. From genome analysis three different unreported clusters were identified. It is also worth noting that almost half of the proteins are annotated as hypothetical protein further demonstrating the novelty associated with this strain. Biological assays on the bacterial extracts confirmed some of the in silico predicted bioactivities. Specifically, strain KRL4 has shown promising anthelmintic activities, and the first report for members of this genus. Our results demonstrate that strain KRL4 could be a promising source of novel bioactive compounds, and the availability of the full genome sequence will enable the identification and recombinant expression of these. This work confirms the promising potential of alpine microorganisms and prompts future efforts aiming at a large-scale exploitation of this resource for natural product discovery.

## Figures and Tables

**Figure 1 microorganisms-09-00890-f001:**
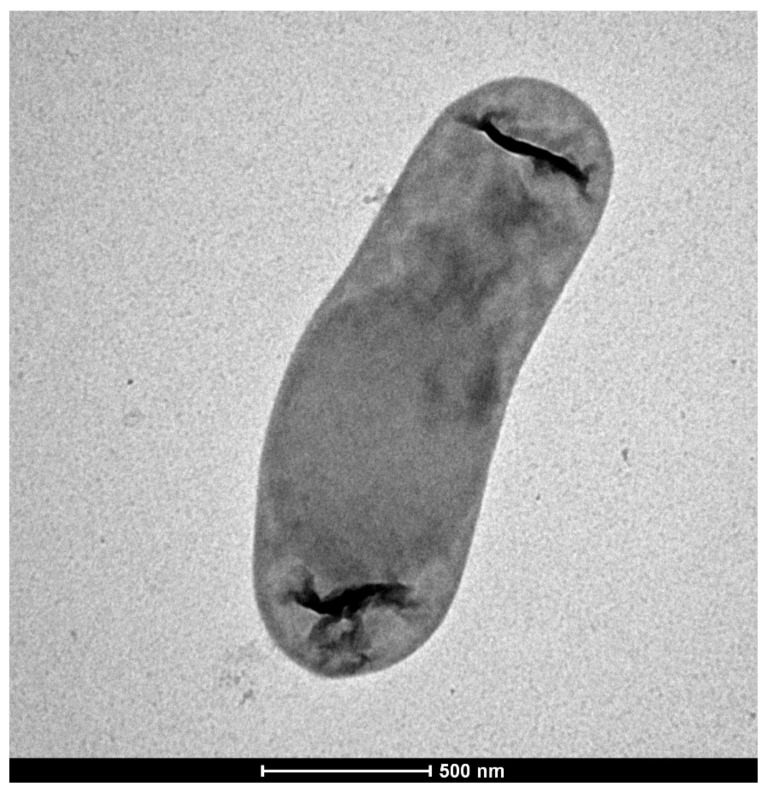
Electron micrograph of a representative KRL4 cell cultured in LB medium. Dark subterminal features may be an artifact of the TEM imaging, however these have been noted in other *Exiguobacterium* isolates but have to date not been explained.

**Figure 2 microorganisms-09-00890-f002:**
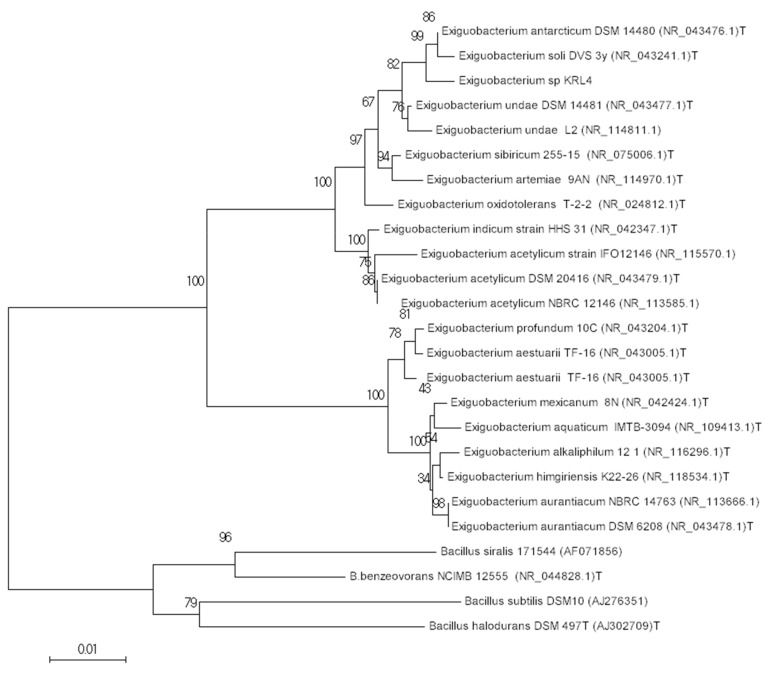
Evolutionary relationships of taxa. The evolutionary history was inferred using the Neighbor-Joining method [57]. The percentage of replicate trees in which the associated taxa clustered together in the bootstrap test (1000 replicates) are shown next to the branches [58]. The tree is drawn to scale, with branch lengths in the same units as those of the evolutionary distances used to infer the phylogenetic tree. The evolutionary distances were computed using the Tamura 3-parameter method [59] and are in the units of the number of base substitutions per site. The analysis involved 25 nucleotide sequences. All positions containing gaps and missing data were eliminated. There was a total of 1333 positions in the final dataset. Evolutionary analyses were conducted in MEGA7 [60].

**Figure 3 microorganisms-09-00890-f003:**
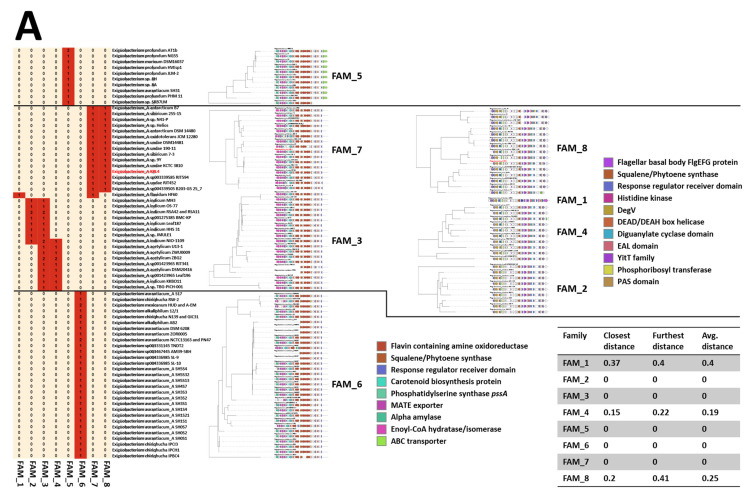
Comparison of terpene (**A**), siderophore (**B**) and (**C**) RiPP pathways identified in *Exiguobacteraceae* including KRL4. Tables show the presence/absence of pathways in *Exiguobacterium* or *Exiguobacterium*_A species while gene cluster families (GCF) are shown in the maps on the right.

**Figure 4 microorganisms-09-00890-f004:**
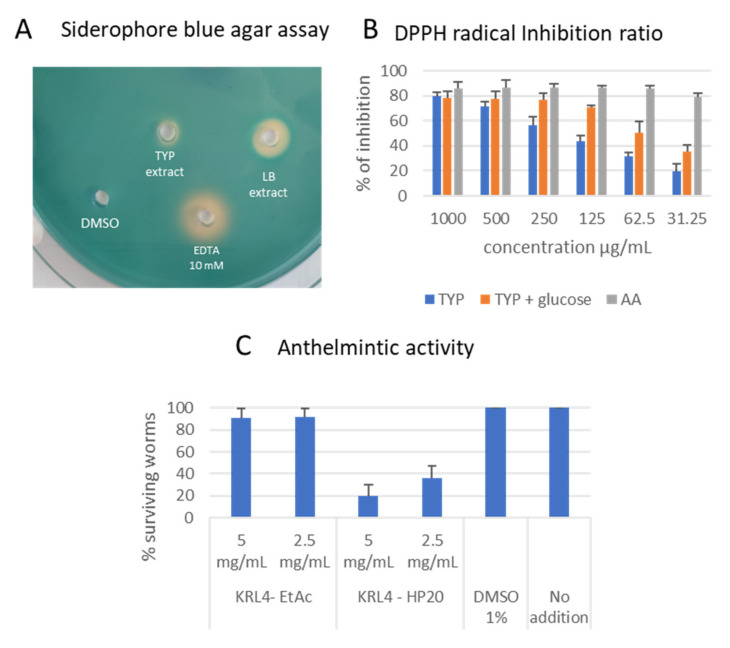
Bioactivity assays. (**A**) Blue agar assays with extracts. (**B**) DPPH radical assay to evaluate antioxidant activities. (**C**) Anthelmintic activity of KRL4 extracts using liquid assay with *C. elegans.*

**Table 2 microorganisms-09-00890-t002:** Genome statistics.

Attribute	Value
Genome size (bp)	3,118,075
DNA coding (bp)	2,823,885
DNA G+C (bp)	1,462,377
DNA scaffolds	48
Total genes	3201
Protein coding genes	3079
RNA genes	79
Pseudo genes	43
Genes in internal clusters	
Genes with function prediction	1522
Genes assigned to COGs	2052
Genes with Pfam domains	2597
Genes with signal peptides	146
Genes with transmembrane helices	1207
CRISPR repeats	3	

## Data Availability

Genomic data are deposed to GenBank under the accession number MOLV00000000.1. See Appendix A for more details.

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
