# Peer review of "Isolation and Characterization of Strain Exiguobacterium sp. KRL4, a Producer of Bioactive Secondary Metabolites from a Tibetan Glacier"

_microorganisms, 2021, doi:10.3390/microorganisms9050890_

Round 1

Reviewer 1 Report

In this paper, the authors isolated and identified the bacterium Exiguobacterium sp. KRL4. A draft genome sequence was obtained and the sequence was analyzed for the presence of biosynthetic clusters for secondary metabolites. The authors then used crude extracts from the organism for various bioassays, and observed antioxidant, siderophore, and nematocidal activities. I recommend the acceptance of the manuscript upon addressing the following comments:

L17 - Indicate that it is in China; i.e., Tibet, China

L48 - vague statement (revise)

L53 - "whose interest" is incorrect, rephrase this section for clarity

L75-83 - incorrect and inconsistent capitalization of reagent names. Yeast, glucose, yeast extract should not be capitalized. The chemical formulae of the hydrates are also presented in the wrong format.

L84 - This statement says that characterization was performed as follows but then, no further info is provided relating to this.

L87 - remove "in" between liquid and M9

L94 - transmission should not be capitalized

L98 - Why was marine agar used?

L117 - 120 - Fix the grammar of this statement

L121 - Replace "has been" with "was"

L130 - "between" is wrongly cut

L165 - Specify what liquid media was used.

L169 and 176 - Specify the unit of time instead of an apostrophe.

L182 - What samples?

L196 - "liquid" should not be capitalized

L198-120 - Wrong capitalization of names

L199 - How was the activity measured?

L225 - "so prepared" needs to be reworded

L226 - EDTA "was"; DMSO "was"

L236 - Incomplete sentence. Revise.

L254-257 - Where are the NMR and LC-MS data. UV data should be shown in the SI.

L256 - Define MK6 and MK7

L272 - Spell out FAMEs

Author Response

Response to Reviewer #1

We thank the reviewer for carefully reading and evaluate our paper. Please find below our answers to his comments. Track-change was activated to allow quick verification of the main text.

In this paper, the authors isolated and identified the bacterium Exiguobacterium sp. KRL4. A draft genome sequence was obtained and the sequence was analyzed for the presence of biosynthetic clusters for secondary metabolites. The authors then used crude extracts from the organism for various bioassays, and observed antioxidant, siderophore, and nematocidal activities. I recommend the acceptance of the manuscript upon addressing the following comments:

L17 - Indicate that it is in China; i.e., Tibet, China

This was indicated in the text.

L48 - vague statement (revise)

The sentence was rephrased as follows: “Among extreme environments, cold environments (< 5° C) are the most widespread on our planet, but so far, applied research on cold-adapted microorganisms was mostly based on the isolation of cold-adapted enzymes”

L53 - "whose interest" is incorrect, rephrase this section for clarity

The sentence was rephrased as follows: “The deep sea represents another cold “niche” that is the subject of a growing interest in the last few years and has yielded a significant number of NPs with different bioactivities.”

L75-83 - incorrect and inconsistent capitalization of reagent names. Yeast, glucose, yeast extract should not be capitalized. The chemical formulae of the hydrates are also presented in the wrong format.

All these remarks were corrected in the text.

L84 - This statement says that characterization was performed as follows but then, no further info is provided relating to this.

The sentence was removed.

L87 - remove "in" between liquid and M9

This was removed suggested.

L94 - transmission should not be capitalized

This was corrected as suggested.

L98 - Why was marine agar used?

Marine agar was not used. This was a mistake and it was corrected.

L117 - 120 - Fix the grammar of this statement

The sentence was rephrased as follows: “High-quality genomic DNA was extracted and purified using a GenElute™ Bacterial Genomic DNA Kit (Sigma, St Louis, MO, USA). The genomic DNA was purified according to manufacturer’s instruction for extraction of nucleic acid from Gram-positive bacteria.”

L121 - Replace "has been" with "was"

This was corrected as suggested.

L130 - "between" is wrongly cut

This was corrected as suggested.

L165 - Specify what liquid media was used.

The media used were indicated.

L169 and 176 - Specify the unit of time instead of an apostrophe.

The unit of time were specified.

L182 - What samples?

The sentence was rephrased for clarity.

L196 - "liquid" should not be capitalized L198-120 - Wrong capitalization of names

These were corrected as suggested.

L199 - How was the activity measured?

A short paragraph was added to define the anthelmintic activity.

L225 - "so prepared" needs to be reworded

The sentence was reworded as follows: “Small holes with a diameter of 1 cm were cut in these dishes and 10 uL of KRL4 extracts were placed inside the holes.”

L226 - EDTA "was"; DMSO "was"

This was corrected as suggested.

L236 - Incomplete sentence. Revise.

The sentence was revised as follows: “Cells have a rod shape, are non-motile and are 2.0-2.5 by 0.5 to 0.8 µm as shown in Figure 1.”

L254-257 - Where are the NMR and LC-MS data. UV data should be shown in the SI.

All these data were added to the SI and correctly cited in the main text.

L256 - Define MK6 and MK7

The definition of MK6 and MK7 was given in the text.

L272 - Spell out FAMEs

The definition of FAMEs was given in M&M (line 173), therefore the acronym was used in this case.

Reviewer 2 Report

The manuscript is interesting and provides new data regarding isolation and characterization of strain Exiguobacterium sp. 2 KRL4 a producer of bioactive secondary metabolites from a Tibetan Glacier. The subject of this manuscript is consistent with the scope of the Journal. The conclusions corresponds with the work's content. These results are interesting and very important for future of biotechnological application.

Manuscript can be published in scientific Microorganisms after some changes (minor revision):

• Line 72-73: Please complete the environmental details on the description of the collection site for strain isolation (pH, EC, ect.). This information is important for the description of the strain isolate.

• Line 93-94:There is no description of the TEM preparation and the observation conditions. Please complete this part of the methods.

• Line 253-254: Please follow the rule to describe "data not shown", or not include this data in manuscript, or show it, for example in the supplementary section.

• Please, be sure that all the references cited in the manuscript are also included in the reference list and vice versa with matching spellings and dates.

Author Response

Response to Reviewer #2

We thank the reviewer for carefully reading and evaluate our paper. Please find below our answers to his comments. Track-change was activated to allow quick verification of the main text.

The manuscript is interesting and provides new data regarding isolation and characterization of strain Exiguobacterium sp. 2 KRL4 a producer of bioactive secondary metabolites from a Tibetan Glacier. The subject of this manuscript is consistent with the scope of the Journal. The conclusions corresponds with the work's content. These results are interesting and very important for future of biotechnological application.

Manuscript can be published in scientific Microorganisms after some changes (minor revision):

  • Line 72-73: Please complete the environmental details on the description of the collection site for strain isolation (pH, EC, ect.). This information is important for the description of the strain isolate.

Unfortunately, we are not in possess of these information. The samples for the isolation was donated to us and these details were not given.

  • Line 93-94:There is no description of the TEM preparation and the observation conditions. Please complete this part of the methods.

The protocol for TEM preparation and observation was included in the methods.

  • Line 253-254: Please follow the rule to describe "data not shown", or not include this data in manuscript, or show it, for example in the supplementary section.

NMR and LC-MS were included in the SI and correctly cited in the main text.

  • Please, be sure that all the references cited in the manuscript are also included in the reference list and vice versa with matching spellings and dates.

The reference list was thoroughly and 3 wrong citations assignments were corrected.